# Less Is More: A Hard Way to Get Potential Dwarfing Hybrid Rootstocks for Valencia Sweet Orange

Danilo Pereira Costa [1,*,†], Eduardo Sanches Stuchi [2,3], Eduardo Augusto Girardi [2], Alécio Souza Moreira [2], Abelmon da Silva Gesteira [2], Mauricio Antonio Coelho Filho [2], Carlos Alberto da Silva Ledo [2], André Luiz Vanucci da Silva [4], Helton Carlos de Leão [5], Orlando Sampaio Passos [2] and Walter dos Santos Soares Filho [2]

[1] Post-Graduation Program in Genetics and Molecular Biology, Universidade Estadual de Santa Cruz—UESC, Rodovia Jorge Amado, Km 16, Salobrinho, Ilhéus 45662-900, Brazil
[2] Embrapa Mandioca e Fruticultura, Rua Embrapa-sn, Cruz das Almas 44380-000, Brazil; eduardo.stuchi@embrapa.br (E.S.S.); eduardo.girardi@embrapa.br (E.A.G.); alecio.moreira@embrapa.br (A.S.M.); abelmon.gesteira@embrapa.br (A.d.S.G.); mauricio-antonio.coelho@embrapa.br (M.A.C.F.); carlos.ledo@embrapa.br (C.A.d.S.L.); orlando.passos@embrapa.br (O.S.P.); walter.soares@embrapa.br (W.d.S.S.F.)
[3] Fundação Coopercitrus-Credicitrus—FCC, Rodovia Brigadeiro Faria Lima, km 384, Bebedouro 14713-000, Brazil
[4] Rua João Pessoa, nº 533, Centro, Matão 15990-020, Brazil; andrevanucci@yahoo.com.br
[5] José Marques Pinheiro Filho, nº 191, Jardim Maracanã, Araraquara 14802-480, Brazil; leao.hc@gmail.com
* Correspondence: danilocosta_1739@hotmail.com; Tel.: +55-759-9210-7465
† Present address: Embrapa Mandioca e Fruticultura, Cruz das Almas 44380-000, Brazil.

**Abstract:** As in several fruit crops, citrus trees with decreased size allow for a higher planting density, which may lead to higher productivity and facilitate operations such as harvesting and spraying. The use of dwarfing rootstocks is one of the most feasible methods for tree size control, but few commercial varieties are available to date. In this work, the long-term performance of Valencia sweet orange grafted onto 51 hybrid citrus rootstocks was evaluated in rainfed cultivation at 6.0 m × 2.5 m tree spacing in Northern São Paulo State, Brazil. About a third of the evaluated hybrids were classified as dwarfing and semi-dwarfing rootstocks, that is, respectively inducing a relative canopy volume of <40% and 40–60% compared with the standard rootstock, the Rangpur lime Santa Cruz selection. The production efficiency and soluble solids concentration were conversely related to the canopy volume. Three citrandarins of Sunki mandarin (TSKC) × Flying Dragon trifoliate orange (TRFD) were grouped within the most productive dwarfing rootstocks. Other hybrids that expressively decreased tree size were mainly sensitive to drought; therefore, the mean fruit yield was low, indicating the need for irrigation, albeit fruit quality was high. Estimated productivity on the selected TSKC × TRFD rootstocks would double to an average of 40 t·ha$^{-1}$·year$^{-1}$ if tree spacing was adjusted to the smaller tree size. Although the HTR-208 citrandarin and the LCR × CTSW-009 citrumelimonia were as vigorous as the Santa Cruz Rangpur lime, they induced an outstanding fruit yield due to their highest tolerance to drought and, hence, can be considered for rainfed cultivation at wider tree spacing.

**Keywords:** *Citrus* spp.; *Poncirus trifoliata*; drought tolerance; fruit quality; fruit yield; graft compatibility; tree size control

## 1. Introduction

Sweet orange is one of the most cultivated fruits in the world, with a total production of around 78.7 million tons in 2019, with Brazil, China, India, USA, and Mexico being the major producers [1]. The consumption of frozen and concentrated orange juice (FCOJ) has been decreasing since the 2000s while that of not from concentrate (NFC) orange juice has increased [2]. Intensive pest and disease management programs are necessary due to the

increasing spread of various limiting pathogens [3,4]. The use of higher tree density has been considered to potentially contribute to augmenting productivity and diminishing costs [5,6]. In São Paulo State, the average tree density increased almost 2-fold, from 362 to 616 trees·ha$^{-1}$, in the 1990–2019 period [7] while the mean productivity increased by 50% [1,7]. In the era of huanglongbing epidemy, higher tree densities were also associated with a lower disease incidence [8] and higher economic feasibility [9].

In high tree density, tree size control becomes necessary to keep trees within the provided space and to allow the movement of equipment aiming at operational efficiency [10]. Moreover, pesticide spraying is usually dosed according to the canopy size [11], which also plays a critical role in harvesting [12]. Citrus dwarfing rootstocks, that is, those which decrease the canopy volume by at least 60% compared with standard-sized rootstocks [13], or those which present a maximum plant height of about 2.4 m [14], are the best low-input tree size control method [15].

In spite of the increasing interest, just a few commercial rootstock varieties have been classified as truly dwarfing ones [16,17]. Indeed, only the Flying Dragon trifoliate orange (*Poncirus trifoliata* var. *monstrosa* (T. Itô) Swingle) has gained such a reputation to date, being reputable as promising in tropical regions [15,18–21], but to a lesser extent under cooler climates and iron-deficient soils [16,22]. In Brazil, the Flying Dragon trifoliate orange has been widely used as the rootstock of the Persian lime (*Citrus ×latifolia* (Yu. Tanaka) Tanaka) [23]. On the other hand, the Flying Dragon presents disadvantages such as low tolerance to drought and graft incompatibility with some scion varieties [24]. All other commercial rootstock varieties in Brazil induce high tree size and, thus, hamper plant management under high tree density because of the intensive pruning to curb overgrowth, which in turn may decrease the yield [5]. Therefore, new productive citrus dwarfing rootstocks are dramatically needed for a wider diversification of varieties, under a scenario of increasing pressure for more sustainable production systems. Most citrus rootstock breeding programs have relied on controlled hybridizations to create genetic diversity for finding suitable rootstocks [15,24], which may include dwarfing rootstocks.

In this work, we evaluated over twelve years the tree size and survival rate, the fruit production and quality, the drought tolerance, and the graft compatibility of Valencia sweet orange (*C. ×sinensis* (L.) Osbeck cv. IAC) on 51 hybrid rootstocks in rainfed cultivation in Northern São Paulo State, Brazil. Three selected citrandarins of Sunki mandarin (*C. sunki* (Hayata) hort. ex Tanaka)× Flying Dragon trifoliate orange induced a mean reduction of 70% in the scion canopy volume compared with the standard rootstock, the Rangpur lime (*C. ×limonia* Osbeck), and their potential use as alternative dwarfing rootstocks is discussed.

## 2. Materials and Methods

### 2.1. Plant Material and Experimental Design

The Valencia sweet orange was used as the scion variety because of its wide cultivation in Brazil and other countries [7,25]. Insect-proof, container-grown nursery trees were grafted onto 51 hybrid citrus rootstocks that were obtained or introduced by the Citrus Breeding Program of Embrapa Cassava and Fruits (Table 1).

The experimental design was in randomized blocks with three replications that corresponded to a planting line each, and five trees in the plot, given a total of 765 trees evaluated. Moreover, the Santa Cruz Rangpur lime and the Tropical Sunki mandarin were included as controls because these two rootstock species are markedly vigorous, drought-tolerant, and commercially used in Brazil [26]. The data from these control rootstocks were used from another trial just next to the experimental area (about 15 m away from each other's center on very uniform soil) [27], with the same nursery tree standard and cultivation conditions, yet being one year older; therefore, results were considered sufficiently comparable by the authors to be added to the analyses.

**Table 1.** Acronyms and corresponding parentals of 51 hybrid citrus rootstocks grafted with Valencia sweet orange (*Citrus ×sinensis* (L.) Osbeck) and evaluated in Northern São Paulo State, Brazil.

| Acronyms [1] and Types of Citrus Hybrid | Parentals |
|---|---|
| HTR-206 citrangedarin | Sunki mandarin (*C. sunki* (Hayata) hort. ex Tanaka)× Coleman citrange (*C. × sinensis × Poncirus trifoliata* (L.) Raf.) |
| HTR-207 citrangedarin | Sunki mandarin × Troyer citrange |
| HTR-208 citrandarin | Sunkimandarin× Benecke trifoliate orange (*P. trifoliata*) |
| LCR × CTSW-009 citrumelimonia | Rangpur lime (*C. × limonia* Osbeck) × Swingle citrumelo (*C. × paradisi* Macfad. ×*P. trifoliata*) |
| LCR × LRF-034 lemon | Rangpur lime × Florida rough lemon (*C. × jambhiri* Lush.) |
| Santa Cruz Rangpur lime | Rangpur lime cv. Santa Cruz |
| LRF × (LCR × TR)-004 citrimoniambhiri | Florida rough lemon × (Rangpur lime × trifoliate orange) |
| LVK × CTSW-009 citrumelemon | Volkamer lemon (*C. ×volkameriana* (Risso) V. Ten. & Pasq.)× Swingle citrumelo |
| MXWL × LHA-001 tangor | Willow leaf mandarin (*C. deliciosa* Ten.) × Hamlin sweet orange |
| Sunki × Alemowa lemandarin | Sunki mandarin × Alemow (*C. macrophylla* Wester) |
| TSKC × (LCR × TR)-016, 020 and 040 citrimoniandarins | Common Sunki mandarin × (Rangpur lime × trifoliate orange) |
| TSKC × CTARG-015, 019, 020, 069 and 081 citrangedarins | Common Sunki mandarin × Argentina citrange |
| TSKC × CTCM-008citrangedarin | Common Sunki mandarin × Coleman citrange |
| TSKC × CTQT 1439-003 and 014 citrangequatandarins | Common Sunki mandarin × Thomasville citrangequat cv. 1439 (*Fortunella margarita* (Lour.) Swingle × Willits citrange) |
| TSKC × CTSW-017, 018, 022, 025, 031, 036, 053, 057 and 058 citrumelandarins | Common Sunki mandarin × Swingle citrumelo |
| TSKC × CTTR-012, 028 and 029 citrangedarins | Common Sunki mandarin × Troyer citrange |
| TSKC × LHA-007 tangor | Common Sunki mandarin × Hamlin sweet orange |
| TSKC × TRBK-006 citrandarin | Common Sunki mandarin × Benecke trifoliate orange |
| TSKC × TRFD-003, 006,and 007 citrandarins | Common Sunki mandarin × Flying Dragon trifoliate orange (*P. trifoliata* var. *monstrosa* (T. Itô) Swingle) |
| TSKFL × CTARG-023 and 029 citrangedarins | Sunki mandarin cv. Florida × Argentina citrange |
| TSKFL × CTC13-005 and 012 citrangedarins | Sunki mandarin cv. Florida × C13 citrange |
| TSKFL × CTSW-004 and 009 citrumelandarins | Sunki mandarin cv. Florida × Swingle citrumelo |
| TSKFL × CTTR-004, 006, 013, 017, 021,and 022 citrangedarins | Sunki mandarin cv. Florida × Troyer citrange |
| Tropical Sunki mandarin | Sunki mandarin cv. Tropical |

[1] Acronyms used by the Citrus Breeding Program of Embrapa Cassava and Fruits with serial numbers identifying each hybrid obtained by the respective cross (that is, indicating different siblings), except for Sunki×Alemow which was introduced from Colombia.

### 2.2. Environmental Conditions and Plant Care

The experiment was initiated by planting in 2008 in a commercial farm located in the municipality of Colômbia, Northern São Paulo, Brazil (20°19′22″ S, 48°41′10″ W, 492 m). Local climate is classified as Aw (tropical savannah with warm rainy summer and dry winter) according to the Köppen classification (Figure 1).

Planting was on a typical deep, dark red oxisol with clayey to medium texture. In 2018, soil attributes at 0–20 cm were pH (CaCl$_2$) = 4.7; Cation Exchange Capacity (CEC) = 51 cmol$_c$·dm$^{-3}$; Ca = 14 cmol$_c$·dm$^{-3}$; Mg = 7 cmol$_c$·dm$^{-3}$; K = 2.1 cmol$_c$·dm$^{-3}$; H + Al = 28 cmol$_c$·dm$^{-3}$, V = 45%; P = 125 mg·dm$^{-3}$; and Organic Matter (OM.) = 14 g·kg$^{-1}$.

Standard cultural practices for the cultivation of sweet orange in the State of São Paulo were followed [28]. The tree spacing was 6.0 m (between-rows) × 2.5 m (in-rows), given 667 trees·ha$^{-1}$ which is representative of the mean tree density used in commercial orchards in the region [7]. There was no complementary irrigation in the evaluation period, and trees were pruned (topping and hedging) in 2018 after harvesting to maintain tree canopy size.Annual mean rates of fertilizers in the evaluation period consisted of 320 g of N, 130 g of P$_2$O$_5$, and 230 g of K$_2$O on a per tree basis, plus 2.2 t·ha$^{-1}$ of limestone. Citrus sudden death (CSD) disease is prevalent [29], while huanglongbing (HLB) is seldomly registered in the region of the experiment [30]. Nevertheless, the Asian citrus psyllid and other pests were rigorously controlled with contact insecticide sprays throughout the evaluation period.

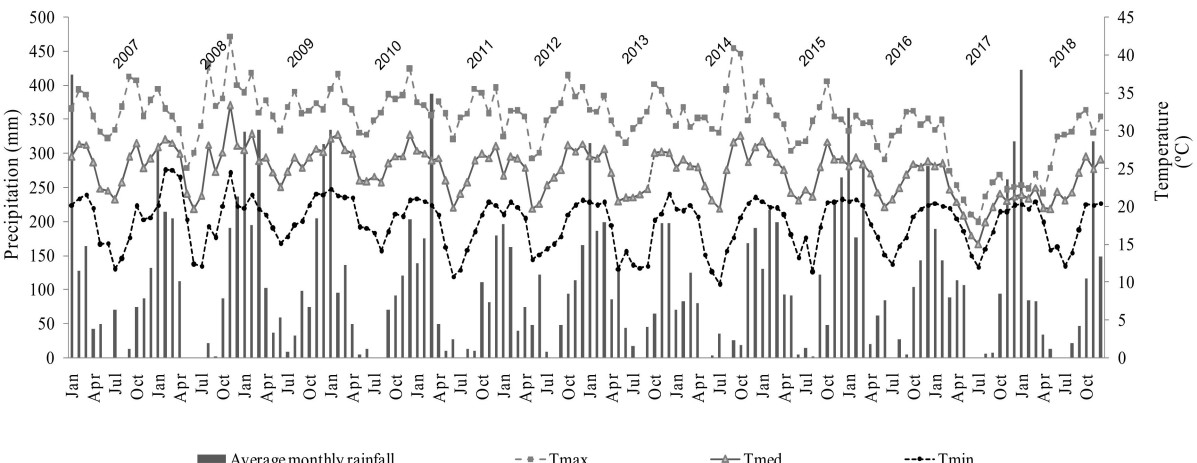

**Figure 1.** Monthly average air temperature (maximum, mean, and minimum) and rainfall in the experimental area from January 2007 to December 2018. Colômbia, São Paulo State, Brazil.

### 2.3. Tree Size

Tree size was assessed by the measurement of tree height (PH (m), from the ground level to the apex) and canopy diameter (on the maximum width in both transversal and longitudinal directions to the planting line, with the mean diameter being later calculated, *CD* (m)), using a ruler. Measurements were performed from 2010 to 2015 in March/April of each year, just after the summer season. The canopy volume (*CV*, m$^3$) was estimated according to [31], Equation (1):

$$CV = \frac{2}{3}\pi (CD/2)^2 \times \text{PH} \tag{1}$$

Only data from 2015 are presented because trees reached full bearing age. Rootstocks were classified according to the induced tree size as described by Castle and Phillips [13]: dwarfing, semi-dwarfing, semi-standard, standard, and super-standard rootstocks led to a relative canopy volume of <40%, 40–60%, 60–80%, 80–100%, and >100% compared with the standard rootstock, in this case the Santa Cruz Rangpur lime.

### 2.4. Fruit Yield, Production Efficiency, Alternate Bearing, and Earliness of Production

Fruits were harvested from 2010 to 2015 in October/November of each year based on the monitoring of the fruit maturation by the grower. Every year, fruits were weighed on a scale, and the average fruit yield (*FY*, kg·tree$^{-1}$) in the period is presented. In 2016 and 2017, the fruit yield was not measured due to operational limitations. In addition, data on fruit yield and tree size were collected from 2018 to 2020, only for 14 rootstocks that were selected as more promising based on their agronomic performance over the 2010–2015 period (production, productive efficiency, fruit quality, tree size, and drought tolerance).

The production efficiency (*EfP*, kg·m$^{-3}$) was calculated in the 2010–2015 period [32], Equation (2):

$$EfP = FY/CV \tag{2}$$

where *FY* = fruit yield, and *CV* = canopy volume in every year from 2010 to 2015, and the average is presented [32].

The alternative bearing index (*ABI*) was calculated by Equation (3):

$$ABI = 1/(n-1) \; x \; \{|(a2-a1)|/(a2+a1) + |(a3-a2)|/(a3+a2) + ... + |(a(n) - a(n-1))|/(a(n) + a(n-1))\} \tag{3}$$

where *n* is the number of evaluated harvests, and *a*1, *a*2, ... , *a*(n− 1), *a*(n) are the fruit production in the respective years [33]. The earliness of production (EA) was estimated by

the relation between the cumulative fruit production of the initial three harvests (2010–2012) and the total cumulative production (2010–2015), expressed in percentage.

### 2.5. Tolerance to Drought, Tree Survival Rate, and Graft Compatibility

The rootstock tolerance to drought (DT) was assessed through a visual scoring of water deficiency symptoms as described by Soares Filho et al. [34]: 1—low tolerance to drought (severe leaf wilting on the whole canopy, leaf drop, and yellowing); 2—moderate tolerance to drought (partial leaf wilting); and 3—high tolerance to drought (absence of leaf wilting and any other visual water deficiency symptoms). Assessment of each tree was carried out independently by two inspectors during the driest month of each year (aiming to find plants tolerant to drought stress) from 2011 to 2017, except in 2013 and 2015 due to operational limitations, and the average value was calculated.

The tree survival rate (SR, %) was calculated by the relation between the cumulative number of dead trees in 2018 and the total number of trees planted per plot of each rootstock evaluated. The graft compatibility (GC) was assessed in November 2017 on 20 hybrid rootstocks selected for their overall good performance to that date. Six plants of each rootstock were assessed. A bark strip (3.0 cm × 5.0 cm) was removed from the graft union region using a switchblade. A visual rating was adapted from Fadel et al. [35]: 1—no symptoms of graft incompatibility; 2—the presence of a fine line between the scion and the rootstock tissues; 3—marked line at the graft union; and 4—sunken lines between the scion and the rootstock tissues with rootstock phloem yellowing and necrosis.

### 2.6. Fruit Quality

The fruit quality was assessed using ten fruits per plot randomly picked from the medium section in the four cardinal directions of the tree canopy. The fruits were assessed just before harvesting from 2010 to 2015, except for 2014, and the mean values are presented. The fruit weight (*FW*, g) was measured on a digital scale. The juice was extracted using a small point-of-sale extractor (Otto 1800, OIC, Limeira, São Paulo, Brazil), and the juice content (*JC*, %) was calculated by the percentual ratio between the juice weight and the fruit weight, Equation (4):

$$JC \ = \ (JW \ \times \ 100)/FW \tag{4}$$

where *JW* = juice weight, and FW = fruit weight, expressed as percentage basis.

The total soluble solids concentration (*SS*, °Brix) in the juice was measured with a digital refractometer (Palette PR-101, ATAGO, Tokyo, Japan), and the values were corrected to 20 °C. The total titratable acidity (*TA*, %) was measured by titration with NaOH (0.3125 N) and phenolphthalein indicator.

The fruit maturity index (*MI*) was calculated by Equation (5):

$$MI \ = \ SS/TA \tag{5}$$

where *SS* = soluble solids, and *TA* = titratable acidity.

The technological index (*TI*, kg·box$^{-1}$) was calculated by Equation (6):

$$TI = [JC \times SS \times 40.8] \times 10000^{-1} \tag{6}$$

where *JC* = juice weight, *SS* = soluble solids, and 40.8 kg is the weight of the standard industrial box of sweet orange.

### 2.7. Statistical Analyses

Data were submitted to variance analyses after statistical assumptions were observed. The means were grouped by the Scott–Knott test ($p < 0.01$, and $p < 0.10$ for *ABI* variable). Linear regression analyses were performed to correlate selected variables. Statistical procedures were carried out using the SAS software [36]. Moreover, multivariate analyses were performed to group the rootstocks based on variables that were classified as the most important for the selection of dwarfing rootstocks by the authors: *FY*, *EfP*, SS, *JC*, DT,

and PH. The genetic distance based on the algorithm of Gower [37] was calculated on the dataset. Principal Component Analysis (PCA) was performed using standardized data, with the auto-values (variance associated with each principal component) being estimated by the characteristic roots of the covariance, and the auto-vectors (set of weighting coefficients of the principal components) being estimated by the elements of the corresponding characteristic vectors [38]. The hierarchic grouping of the individual and simultaneous analyses was obtained from the genetic distance matrices using the Unweighted Pair Group Method with Arithmetic Mean (UPGMA) method [39]. The grouping validation was determined through the Cophenetic Correlation Coefficient (CCC) [40]. The CCC significance was calculated by the Mantels *t*-test using 1000 permutations [41]. The genetic distance matrices by individual analyses and the CCC were calculated using the Genes software [42]. The genetic distance matrix based on Gower's algorithm was calculated using the R software [43]. The dissimilarity dendrogram was obtained using the Statistica 7.1 software [44]. The criterion used to determine the ideal number of groups was the pseudo-t2 [45] using the "NbClust" package of the R computational software [46].

## 3. Results

### 3.1. Tree Size

The evaluated rootstocks were grouped within five classes of canopy diameter and four classes of plant height, comprising a range from 2.19 m to 4.14 m and from 2.05 m to 3.86 m, respectively, in 2015. As a result, four groups of decreasing canopy volume were formed, with averages of 26.8, 19.5, 13.1, and 7.8 m$^3$ (Table 2). Within each hybrid rootstock, evaluated trees presented very uniform tree size across all replications.

**Table 2.** Tree canopy diameter (*CD*) and height (*PH*), and canopy volume (*CV*) in 2015, and mean fruit yield (*FY*) and production efficiency (*EfP*) in the 2010–2015 period of Valencia sweet orange (*Citrus×sinensis* (L.) Osbeck) grafted onto 51 rootstocks in Northern São Paulo State, Brazil.

| Rootstock | *CD* | | *PH* | | *CV* (m$^3$) | | *FY* (kg·tree$^{-1}$) | | *EfP* (kg·m$^{-3}$) | |
|---|---|---|---|---|---|---|---|---|---|---|
| | ——— (m) ——— | | | | | | | | | |
| HTR-206 citrangedarin | 3.90 | a | 3.57 | a | 28.81 | a | 45.10 | b | 3.02 | d |
| HTR-207 citrangedarin | 2.73 | d | 2.74 | c | 10.71 | c | 25.38 | d | 4.78 | c |
| HTR-208 citrandarin | 4.02 | a | 3.35 | b | 28.47 | a | 68.22 | a | 4.09 | c |
| LCR × CTSW-009 citrumelimonia | 3.57 | b | 3.36 | b | 22.63 | b | 42.72 | b | 4.33 | c |
| LCR × LRF-034 lemon | 3.63 | b | 3.22 | b | 22.31 | b | 39.20 | b | 3.57 | c |
| Santa Cruz Rangpur lime | 3.78 | a | 3.59 | a | 26.94 | a | 39.46 | b | 3.41 | c |
| LRF × (LCR × TR)-004 citrimoniambhiri | 3.48 | b | 2.83 | c | 18.31 | b | 41.82 | b | 3.69 | c |
| LVK × CTSW-009 citrumelemon | 3.46 | b | 3.25 | b | 20.37 | b | 36.07 | b | 2.76 | d |
| MXWL × LHA-001 tangor | 4.01 | a | 3.47 | a | 29.46 | a | 45.12 | b | 2.88 | d |
| Sunki × Alemow alemandarin | 2.86 | c | 2.60 | c | 11.16 | c | 29.92 | c | 4.62 | c |
| TSKC × (LCR × TR)-016 citrimoniandarin | 2.61 | d | 2.26 | d | 8.19 | d | 20.34 | d | 4.36 | c |
| TSKC × (LCR × TR)-020 citrimoniandarin | 3.77 | a | 3.17 | b | 23.52 | a | 38.55 | b | 2.98 | d |
| TSKC × (LCR × TR)-040 citrimoniandarin | 3.57 | b | 3.18 | b | 21.24 | b | 42.20 | b | 3.64 | c |
| TSKC × CTARG-015 citrangedarin | 3.27 | b | 3.48 | a | 19.49 | b | 33.29 | c | 2.94 | d |
| TSKC × CTARG-019 citrangedarin | 3.80 | a | 3.45 | a | 26.25 | a | 35.10 | b | 2.67 | d |
| TSKC × CTARG-020 citrangedarin | 4.14 | a | 3.86 | a | 34.63 | a | 41.82 | b | 2.30 | d |
| TSKC × CTARG-069 citrangedarin | 3.24 | b | 3.12 | b | 17.31 | b | 39.38 | b | 3.71 | c |
| TSKC × CTARG-081 citrangedarin | 3.89 | a | 3.48 | a | 27.54 | a | 34.79 | b | 2.43 | d |
| TSKC × CTCM-008 citrangedarin | 2.63 | d | 2.30 | d | 9.18 | d | 19.92 | d | 5.65 | b |
| TSKC × CTQT 1439-003 citrangequatandarin | 3.41 | b | 3.15 | b | 19.29 | b | 38.32 | b | 3.89 | c |
| TSKC × CTQT 1439-014 citrangequatandarin | 3.35 | b | 3.03 | b | 18.11 | b | 50.18 | b | 4.51 | c |
| TSKC × CTSW-017 citrumelandarin | 3.87 | a | 3.62 | a | 28.55 | a | 28.55 | c | 1.79 | d |
| TSKC × CTSW-018 citrumelandarin | 3.61 | b | 3.47 | a | 24.67 | a | 30.66 | c | 2.55 | d |
| TSKC × CTSW-022 citrumelandarin | 2.94 | c | 2.82 | c | 12.72 | c | 24.87 | d | 3.49 | c |

**Table 2.** *Cont.*

| Rootstock | CD | | PH | | CV (m³) | | FY (kg·tree⁻¹) | | EfP (kg·m⁻³) | |
|---|---|---|---|---|---|---|---|---|---|---|
| | ——— (m) ——— | | | | | | | | | |
| TSKC × CTSW-025 citrumelandarin | 3.36 | b | 2.93 | c | 17.30 | b | 40.31 | b | 4.50 | c |
| TSKC × CTSW-031 citrumelandarin | 2.86 | b | 2.90 | c | 12.85 | c | 26.17 | d | 3.46 | c |
| TSKC × CTSW-036 citrumelandarin | 3.35 | b | 3.18 | b | 18.79 | b | 34.08 | c | 3.08 | d |
| TSKC × CTSW-053 citrumelandarin | 3.51 | b | 3.10 | b | 20.04 | b | 38.45 | b | 3.05 | d |
| TSKC × CTSW-057 citrumelandarin | 3.55 | b | 3.25 | b | 21.43 | b | 37.51 | b | 2.42 | d |
| TSKC × CTSW-058 citrumelandarin | 2.19 | e | 2.08 | d | 5.31 | d | 23.25 | d | 8.11 | a |
| TSKC × CTTR-012 citrangedarin | 3.63 | b | 3.43 | a | 23.80 | a | 35.02 | c | 3.21 | d |
| TSKC × CTTR-028 citrangedarin | 3.35 | b | 3.12 | b | 18.36 | b | 19.65 | d | 2.37 | d |
| TSKC × CTTR-029 citrangedarin | 3.17 | c | 2.96 | c | 15.49 | c | 30.79 | c | 4.33 | c |
| TSKC × LHA-007 tangor | 3.58 | b | 3.52 | a | 23.68 | a | 36.73 | b | 2.71 | d |
| TSKC × TRBK-006 citrandarin | 2.64 | d | 2.32 | d | 8.51 | d | 32.52 | c | 5.27 | b |
| TSKC × TRFD-003 citrandarin | 2.87 | c | 2.67 | c | 11.57 | c | 39.32 | b | 6.41 | b |
| TSKC × TRFD-006 citrandarin | 2.43 | e | 2.05 | d | 6.34 | d | 32.16 | c | 7.68 | a |
| TSKC × TRFD-007 citrandarin | 2.58 | d | 2.27 | d | 8.01 | d | 27.92 | c | 6.70 | a |
| TSKFL × CTARG-023 citrangedarin | 3.36 | b | 3.13 | b | 18.63 | b | 38.49 | b | 3.73 | c |
| TSKFL × CTARG-029 citrangedarin | 3.44 | b | 3.00 | b | 18.63 | b | 40.15 | b | 3.72 | c |
| TSKFL ×CTC13-005 citrangedarin | 3.09 | c | 2.82 | c | 14.25 | c | 29.80 | c | 4.02 | c |
| TSKFL × CTC13-012 citrangedarin | 3.37 | b | 3.29 | b | 19.55 | b | 36.19 | b | 2.86 | d |
| TSKFL × CTSW-004 citrumelandarin | 3.80 | a | 3.16 | b | 23.81 | a | 47.64 | b | 4.23 | c |
| TSKFL × CTSW-009 citrumelandarin | 3.77 | a | 3.63 | a | 27.05 | a | 33.54 | c | 2.46 | d |
| TSKFL × CTTR-004 citrangedarin | 3.28 | b | 3.34 | b | 18.93 | b | 29.90 | c | 2.76 | d |
| TSKFL × CTTR-006 citrangedarin | 2.94 | c | 2.88 | c | 12.96 | c | 20.22 | d | 3.20 | d |
| TSKFL × CTTR-013 citrangedarin | 3.45 | b | 3.14 | b | 19.99 | b | 37.75 | b | 3.24 | d |
| TSKFL × CTTR-017 citrangedarin | 2.77 | d | 2.39 | d | 9.56 | d | 21.15 | d | 5.08 | b |
| TSKFL × CTTR-021 citrangedarin | 3.06 | c | 2.94 | c | 14.49 | c | 27.68 | c | 3.23 | d |
| TSKFL × CTTR-022 citrangedarin | 3.18 | c | 2.82 | c | 14.94 | c | 26.25 | d | 3.59 | c |
| Tropical Sunki mandarin | 3.64 | b | 3.73 | a | 25.97 | a | 41.60 | b | 2.72 | d |
| F | * | | * | | * | | * | | * | |
| CV (%) | 6.80 | | 7.62 | | 18.42 | | 16.53 | | 21.86 | |
| Mean | 3.33 | | 3.07 | | 18.83 | | 34.81 | | 3.77 | |

Means followed by the same letters in the column belong to the same group by the Scott–Knott test. (ns) not significant, (*) significant at *p*<0.01. Hybrid rootstock acronyms are described in Table 1.

According to the tree size classification proposed by Castle and Phillips [13], the rootstocks were classified as super-standard (13.7%), standard (19.7%), semi-standard (33.3%), semi-dwarfing (17.6%), and dwarfing (15.7%) compared with the Santa Cruz Rangpur lime. Only HTR-207, TSKFL × CTTR-017 and TSKC × CTCM-008 citrangedarins, TSKC × TRBK-006, TSKC × TRFD-006 and TSKC × TRFD-007 citrandarins, the TSKC × (LCR × TR)-016 citrimoniandarin, and the TSKC × CTSW-058 citrumelandarin could be highlighted as true dwarfing ones due to inducing tree height from 2.05 m to 2.74 m and a decrease in the canopy volume from 64% to 80% until eight years after planting.

### 3.2. Fruit Yield, Production Efficiency, Alternate Bearing, and Earliness of Production

In 2010–2015, the HTR-208 citrandarin induced the highest mean fruit yield in the Valencia sweet orange, 1.73 times higher than those of trees grafted onto the Santa Cruz Rangpur lime. About half of the evaluated rootstocks induced similar mean fruit yield to the standard rootstock, with most of them being vigorous, except for the TSKC × TRFD-003 citrandarin. Although a lower fruit yield was directly related to a lower tree size, three dwarfing citrandarins (TSKC × TRBK-006 and TSKC × TRFD-006 and -007) were within a group of slightly higher production, with a mean of 30.8 kg·tree⁻¹.

The production efficiency (*EfP*) is a key variable to select citrus rootstocks because it is related to the more efficient use of resources. Four groups were formed presenting a mean *EfP* of, respectively, 7.49, 5.60, 3.98, and 2.76 kg·m⁻³. Only three rootstocks led to the highest *EfP*, TSKC × TRFD-006 and 007, and TSKC × CTSW-058, which were markedly

dwarfing. The Santa Cruz Rangpur lime was within the third group, which comprised 40% of the evaluated rootstocks. This group can be highlighted because it comprised mainly the vigorous rootstocks that conjugated higher fruit yields, whereas about 45% of the evaluated rootstocks presented an even lower *EfP* and, therefore, are of less interest regardless of the tree size classification (Table 2).

Regarding the earliness of production, half of the evaluated rootstocks resulted later bearing in comparison to that of Santa Cruz Rangpur lime. There was no relation between the earliness and other traits, such as tree size, fruit yield, or tolerance to drought. Similar behavior was observed for the alternate bearing, which was generally low. The Santa Cruz Rangpur lime was included among those with a lower *ABI*, from 0.26 to 0.37 (Table 3).

**Table 3.** Tolerance to drought (DT) in the 2011–2017 period except for 2013 and 2015, tree survival rate (SR) in 2018, alternate bearing index (*ABI*) and earliness of production (EA) in the 2010–2015 period, and graft compatibility (GC) in 2017 of Valencia sweet orange (*Citrus ×sinensis* (L.) Osbeck) grafted onto 51 rootstocks in Northern São Paulo State, Brazil.

| Rootstock | DT [1] | | SR | | *ABI*[2] | | EA [3] | | GC [4] (%) | | | |
|---|---|---|---|---|---|---|---|---|---|---|---|---|
| | | | (%) | | | | (%) | | 1 | 2 | 3 | 4 |
| HTR-206 citrangedarin | 1.77 | b | 93.33 | a | 0.38 | a | 34.15 | b | 0 | 100 | 0 | 0 |
| HTR-207 citrangedarin | 1.42 | c | 80.00 | a | 0.34 | b | 30.32 | b | - | - | - | - |
| HTR-208 citrandarin | 2.21 | a | 93.33 | a | 0.38 | a | 28.00 | b | 0 | 100 | 0 | 0 |
| LCR × CTSW-009 citrumelimonia | 2.44 | a | 100.00 | a | 0.39 | a | 40.48 | a | 0 | 100 | 0 | 0 |
| LCR × LRF-034 lemon | 2.09 | a | 100.00 | a | 0.34 | b | 41.66 | a | - | - | - | - |
| Santa Cruz Rangpur lime | 2.04 | a | 46.67 | b | 0.31 | b | 47.53 | a | 0 | 100 | 0 | 0 |
| LRF × (LCR × TR)-004 citrimoniambhiri | 1.56 | c | 86.67 | a | 0.37 | b | 28.08 | b | 0 | 83 | 17 | 0 |
| LVK × CTSW-009 citrumelemon | 1.55 | c | 46.67 | b | 0.40 | a | 23.26 | b | - | - | - | - |
| MXWL × LHA-001 tangor | 2.06 | a | 100.00 | a | 0.36 | b | 44.32 | a | 0 | 100 | 0 | 0 |
| Sunki × Alemow alemandarin | 1.32 | c | 80.00 | a | 0.30 | b | 37.60 | a | - | - | - | - |
| TSKC × (LCR × TR)-016 citrimoniandarin | 1.82 | b | 100.00 | a | 0.40 | a | 21.27 | b | - | - | - | - |
| TSKC × (LCR × TR)-020 citrimoniandarin | 1.57 | c | 86.67 | a | 0.32 | b | 41.58 | a | 33 | 67 | 0 | 0 |
| TSKC × (LCR × TR)-040 citrimoniandarin | 2.15 | a | 100.00 | a | 0.31 | b | 47.36 | a | 0 | 100 | 0 | 0 |
| TSKC × CTARG-015 citrangedarin | 1.69 | c | 93.33 | a | 0.39 | a | 43.74 | a | - | - | - | - |
| TSKC × CTARG-019 citrangedarin | 1.83 | b | 86.67 | a | 0.37 | b | 41.39 | a | 0 | 100 | 0 | 0 |
| TSKC × CTARG-020 citrangedarin | 1.58 | c | 100.00 | a | 0.33 | b | 37.56 | a | 0 | 100 | 0 | 0 |
| TSKC × CTARG-069 citrangedarin | 1.19 | c | 86.67 | a | 0.44 | a | 31.98 | b | - | - | - | - |
| TSKC × CTARG-081 citrangedarin | 1.64 | c | 86.67 | a | 0.29 | b | 40.99 | a | - | - | - | - |
| TSKC × CTCM-008 citrangedarin | 1.54 | c | 93.33 | a | 0.41 | a | 48.09 | a | - | - | - | - |
| TSKC × CTQT 1439-003 citrangequatandarin | 2.00 | a | 93.33 | a | 0.39 | a | 43.09 | a | 0 | 100 | 0 | 0 |
| TSKC × CTQT 1439-014 citrangequatandarin | 2.09 | a | 100.00 | a | 0.43 | a | 31.50 | b | 0 | 100 | 0 | 0 |
| TSKC × CTSW-017 citrumelandarin | 1.36 | c | 20.00 | b | 0.45 | a | 30.87 | b | - | - | - | - |
| TSKC × CTSW-018 citrumelandarin | 1.31 | c | 86.67 | a | 0.46 | a | 38.76 | a | - | - | - | - |
| TSKC × CTSW-022 citrumelandarin | 1.43 | c | 100.00 | a | 0.35 | b | 35.63 | b | - | - | - | - |
| TSKC × CTSW-025 citrumelandarin | 1.91 | b | 73.33 | a | 0.41 | a | 42.21 | a | 0 | 100 | 0 | 0 |
| TSKC × CTSW-031 citrumelandarin | 2.13 | a | 73.33 | a | 0.42 | a | 33.05 | b | - | - | - | - |
| TSKC × CTSW-036 citrumelandarin | 1.87 | b | 80.00 | a | 0.38 | a | 24.57 | b | - | - | - | - |
| TSKC × CTSW-053 citrumelandarin | 1.49 | c | 93.33 | a | 0.45 | a | 28.99 | b | - | - | - | - |
| TSKC × CTSW-057 citrumelandarin | 1.40 | c | 80.00 | a | 0.36 | b | 30.29 | b | - | - | - | - |
| TSKC × CTSW-058 citrumelandarin | 1.81 | b | 40.00 | b | 0.37 | b | 38.85 | a | - | - | - | - |
| TSKC × CTTR-012 citrangedarin | 1.53 | c | 80.00 | a | 0.26 | b | 43.63 | a | - | - | - | - |
| TSKC × CTTR-028 citrangedarin | 1.30 | c | 100.00 | a | 0.34 | b | 39.90 | a | - | - | - | - |
| TSKC × CTTR-029 citrangedarin | 1.48 | c | 100.00 | a | 0.35 | b | 33.72 | b | - | - | - | - |

Table 3. *Cont.*

| Rootstock | DT [1] | | SR (%) | | *ABI* [2] | | EA [3] (%) | | GC [4] (%) | | | |
|---|---|---|---|---|---|---|---|---|---|---|---|---|
| | | | | | | | | | 1 | 2 | 3 | 4 |
| TSKC × LHA-007 tangor | 1.81 | b | 80.00 | a | 0.42 | a | 31.22 | b | - | - | - | - |
| TSKC × TRBK-006 citrandarin | 1.61 | c | 100.00 | a | 0.48 | a | 27.95 | b | 0 | 83 | 0 | 17 |
| TSKC × TRFD-003 citrandarin | 2.03 | a | 100.00 | a | 0.38 | a | 45.13 | a | 0 | 100 | 0 | 0 |
| TSKC × TRFD-006 citrandarin | 1.51 | c | 73.33 | a | 0.29 | b | 43.01 | a | 0 | 100 | 0 | 0 |
| TSKC × TRFD-007 citrandarin | 1.80 | b | 93.33 | a | 0.39 | a | 31.68 | b | 66 | 17 | 17 | 0 |
| TSKFL × CTARG-023 citrangedarin | 1.53 | c | 86.67 | a | 0.28 | b | 30.83 | b | - | - | - | - |
| TSKFL × CTARG-029 citrangedarin | 2.01 | a | 100.00 | a | 0.40 | a | 35.62 | b | 33 | 67 | 0 | 0 |
| TSKFL × CTC13-005 citrangedarin | 1.60 | c | 93.33 | a | 0.42 | a | 35.77 | b | - | - | - | - |
| TSKFL × CTC13-012 citrangedarin | 1.28 | c | 53.33 | b | 0.46 | a | 30.41 | b | - | - | - | - |
| TSKFL × CTSW-004 citrumelandarin | 1.78 | b | 100.00 | a | 0.31 | b | 33.50 | b | 0 | 100 | 0 | 0 |
| TSKFL × CTSW-009 citrumelandarin | 1.55 | c | 100.00 | a | 0.46 | a | 34.18 | b | - | - | - | - |
| TSKFL × CTTR-004 citrangedarin | 1.21 | c | 53.33 | b | 0.35 | b | 40.66 | a | - | - | - | - |
| TSKFL × CTTR-006 citrangedarin | 1.29 | c | 60.00 | b | 0.36 | b | 56.55 | a | - | - | - | - |
| TSKFL × CTTR-013 citrangedarin | 1.66 | c | 73.33 | a | 0.28 | b | 28.80 | b | - | - | - | - |
| TSKFL × CTTR-017 citrangedarin | 1.77 | b | 100.00 | a | 0.33 | b | 49.70 | a | - | - | - | - |
| TSKFL × CTTR-021 citrangedarin | 1.41 | c | 80.00 | a | 0.31 | b | 45.98 | a | - | - | - | - |
| TSKFL × CTTR-022 citrangedarin | 1.49 | c | 100.00 | a | 0.28 | b | 39.90 | a | - | - | - | - |
| Tropical Sunki mandarin | 1.97 | a | 86.67 | a | 0.56 | a | 44.39 | a | 17 | 83 | 0 | 0 |
| F | * | | * | | ** | | * | | | | | |
| *CV* (%) | 11.90 | | 21.96 | | 18.26 | | 24.71 | | | | | |
| Mean | 1.68 | | 84.58 | | 0.37 | | 37.05 | | | | | |

Means followed by the same letters in the column belong to the same group by the Scott–Knott test. (ns) not significant, (*) significant at $p<0.01$, and (**) significant at $p < 0.10$. Hybrid rootstock acronyms are described in Table 1. [1] DT was evaluated as described by Soares Filho et al. [34]. [2] *ABI* was calculated according to Pearce and Dobersek [33]. [3] Cumulative fruit yield in 2010–2012 by the cumulative fruit yield in 2019–2015. [4] GC was adapted from Fadel et al. [35]: 1—no symptoms of graft incompatibility; 2—the presence of a fine line between the scion and the rootstock tissues; 3—marked line at the graft union; and 4—sunken lines between the scion and the rootstock tissues with rootstock phloem yellowing and necrosis. (-) not evaluated.

The canopy volume was directly related to the mean fruit yield per tree in the 2010–2015 period (Figure 2A), that is, larger trees induced by vigorous rootstocks produced more fruits. On the other hand, the production efficiency (kg of fruits per cubic meter of canopy) was conversely related to the canopy volume (Figure 2B). Because there was no significant relation between efficiency and total fruit load (Figure 2C), it is clear that the highest fruit set per cubic meter of the scion canopy could be given only by the most dwarfing rootstocks. Interestingly, the mean concentration of soluble solids in fruits of Valencia sweet orange was also conversely related to the canopy volume (Figure 2D). Juice content and soluble solids concentration are directly subjected to variation due to the osmotic adjustment induced by the rootstock in citrus crops [47]. However, in this work, there was no significant relationship between the rootstock tolerance to drought and the canopy volume (Figure 2E), suggesting that the improvement of juice quality was regulated by other mechanisms rather than only drought-response in dwarfing rootstocks. Since the higher the mean fruit yield, the lower the soluble solids concentration in the juice (Figure 2F), both fruit quality and fruit set might also have benefited from the dwarfed tree growth habit, which allows a more uniform interception of the solar radiation within the small canopy, and also a lower number of fruits and other sinks competing for the available photoassimilates [48–50].

### 3.3. Tolerance to Drought, Tree Survival Rate, and Graft Compatibility

Ten hybrids were the most drought-tolerant rootstocks, being equivalent to the commercial standards, Tropical Sunki mandarin and Santa Cruz Rangpur lime. Another ten hybrids induced intermediate tolerance to drought, and the rest, 57% of the evaluated hybrids, were very sensitive to drought. Although tolerance to drought was not related to

the tree size, most semi-dwarfing and dwarfing rootstocks were intolerant regardless of their fruit yield, yet the TSKC × TRFD-003 citrandarin was as drought-tolerant as the Santa Cruz Rangpur lime. Among the more vigorous rootstocks, only high-bearing hybrids were also drought-tolerant ones (Table 3).

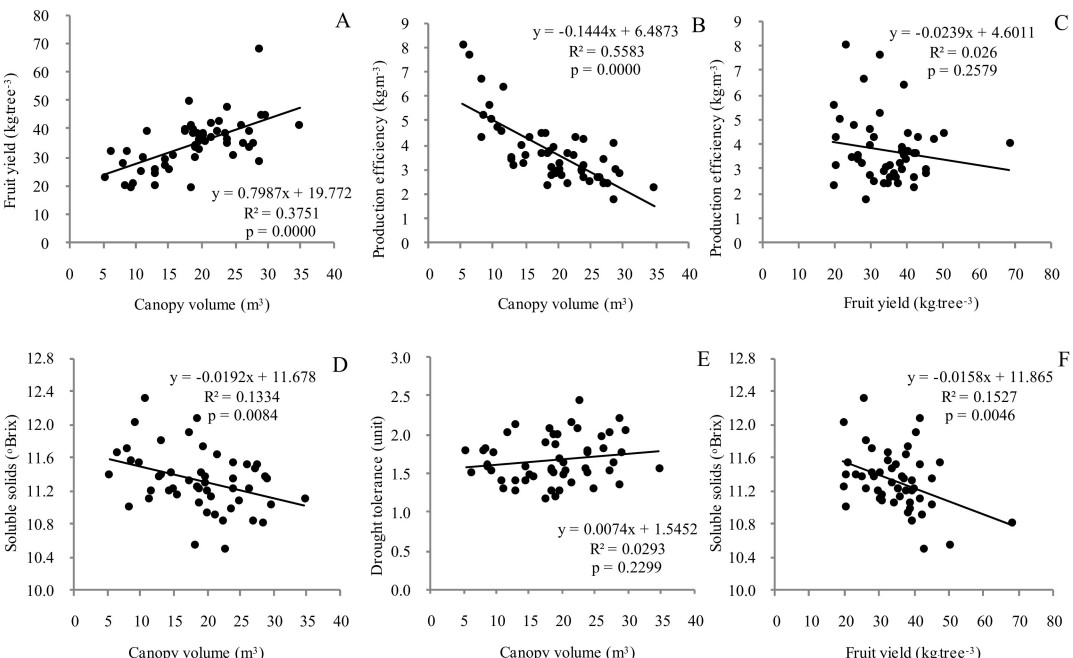

**Figure 2.** Relations between the canopy volume and the mean fruit yield (**A**), the mean production efficiency (**B**), the mean soluble solids concentration (**D**), and the drought tolerance (**E**), and between the mean fruit yield and the mean production efficiency (**C**) and the mean soluble solids concentration (**F**) of Valencia sweet orange (*Citrus×sinensis* (L.) Osbeck) grafted onto 51 rootstocks in the 2010–2015 period in Northern São Paulo State, Brazil.

Almost 50% of the evaluated rootstocks had a mean tree survival rate equal to or higher than 90% ten years after planting, with no tree death for 17 hybrids (about 30% of the total hybrids). On the other hand, Santa Cruz Rangpur lime and six hybrid rootstocks led to an average tree loss of 45%. The graft incompatibility was probably not a cause, because most of the selected rootstocks did not present any symptoms on the graft union (Table 3). Despite not presenting necrosis nor sunken line at the graft union, the TSKC × CTSW-058 citrumelandarin presented other incompatibility symptoms, such as overgrowth of the scion trunk over the rootstock, general tree yellowing, and stunting. This probably explains its low tree survival rate and even the high production efficiency due to a progressive girdling-like effect and decreased tree size.

In respect to other putative causes of tree loss, until twelve years after planting it could be highlighted that: (1) typical visual symptoms of CSD (tree fast decline, rootstock phloem yellowing) were observed only in trees on the Santa Cruz Rangpur lime; (2) a few trees grafted onto Santa Cruz Rangpur lime, Tropical Sunki mandarin, HTR-206 citrangedarin, and MXWL × LHA-001 tangor presented visual symptoms of *Phytophthora* spp. crown rot; (3) a decreasing incidence of visual blight symptoms was observed in some trees of HTR-208 citrandarin > Santa Cruz Rangpur lime > HTR-207 citrangedarin > TSKC × CTSW-025 citrumelandarin; (4) no citrus tristeza virus (CTV)-induced quick decline symptoms were observed in the experimental area; (5) dwarfing hybrids TSKC × TRFD-006 citrandarin and TSKC × CTCM-008 citrangedarin were impaired by severe drought conditions, presenting heavy leaf drop, stem dieback, and some tree death. Notwithstanding, trees grafted onto the former hybrid normally recovered after rainfall, resulting in high flowering and fruit set, whilst trees grafted onto the latter presented lower bearing and some yellowing year-round.

*3.4. Fruit Quality*

Larger fruits were produced by trees on 55% of the evaluated rootstocks including both commercial standards. Most rootstocks also induced the highest juice content (JC), from 45% to 49%, which is within the usual range for sweet orange fruits using a small point-of-sale extractor. Conversely, the TSKFL × CTC13-012 citrangedarin induced the lowest JC, of only 39%. About half of the dwarfing and semi-dwarfing hybrids led to similar JC to the Santa Cruz Rangpur lime (Table 4).

**Table 4.** Fruit weight (FW), juice content (JC), total soluble solids concentration (SS), titratable acidity (TA), maturity index (MI), and technological index (TI) in the 2010–2015 period, except for 2014, of fruits of Valencia sweet orange (*Citrus ×sinensis* (L.) Osbeck) grafted onto 51 rootstocks in Northern São Paulo State, Brazil.

| Rootstock | FW (g) | | JC (%) | | SS (°Brix) | | TA (%) | | MI (SS/TA) | | TI (kg SS box$^{-1}$) | |
|---|---|---|---|---|---|---|---|---|---|---|---|---|
| HTR-206 citrangedarin | 165.36 | b | 46.10 | a | 11.34 | b | 0.66 | b | 17.36 | a | 2.12 | a |
| HTR-207 citrangedarin | 185.13 | a | 42.92 | b | 12.33 | a | 0.82 | a | 16.36 | b | 2.16 | a |
| HTR-208 citrandarin | 205.94 | a | 46.59 | a | 10.81 | b | 0.73 | a | 14.94 | b | 2.06 | b |
| LCR × CTSW-009 citrumelimonia | 193.91 | a | 47.04 | a | 10.51 | b | 0.65 | b | 16.65 | b | 2.02 | b |
| LCR × LRF-034 lemon | 177.53 | b | 48.58 | a | 10.85 | b | 0.66 | b | 16.95 | a | 2.18 | a |
| Santa Cruz Rangpur lime | 202.85 | a | 49.07 | a | 10.85 | b | 0.63 | b | 16.70 | b | 2.09 | b |
| LRF × (LCR × TR)-004 citrimoniambhiri | 162.85 | b | 45.85 | a | 12.08 | a | 0.65 | b | 19.20 | a | 2.24 | a |
| LVK × CTSW-009 citrumelemon | 179.46 | b | 46.05 | a | 11.14 | b | 0.71 | a | 16.11 | b | 2.09 | b |
| MXWL × LHA-001 tangor | 191.83 | a | 45.78 | a | 11.03 | b | 0.69 | b | 15.85 | b | 2.03 | b |
| Sunki × Alemow alemandarin | 176.01 | b | 43.71 | b | 11.12 | b | 0.65 | b | 17.50 | a | 1.99 | b |
| TSKC × (LCR × TR)-016 citrimoniandarin | 205.97 | a | 44.42 | b | 11.01 | b | 0.65 | b | 17.25 | a | 2.02 | b |
| TSKC × (LCR × TR)-020 citrimoniandarin | 175.07 | b | 44.03 | b | 10.99 | b | 0.67 | b | 17.18 | a | 1.99 | b |
| TSKC × (LCR × TR)-040 citrimoniandarin | 197.89 | a | 45.47 | a | 10.91 | b | 0.68 | b | 16.60 | b | 2.05 | b |
| TSKC × CTARG-015 citrangedarin | 177.73 | b | 47.88 | a | 11.31 | b | 0.72 | a | 16.26 | b | 2.22 | a |
| TSKC × CTARG-019 citrangedarin | 192.23 | a | 43.94 | b | 11.22 | b | 0.68 | b | 17.27 | a | 2.02 | b |
| TSKC × CTARG-020 citrangedarin | 174.74 | b | 46.47 | a | 11.12 | b | 0.74 | a | 15.47 | b | 2.09 | b |
| TSKC × CTARG-069 citrangedarin | 180.72 | b | 44.27 | b | 11.33 | b | 0.75 | a | 15.65 | b | 2.07 | b |
| TSKC × CTARG-081 citrangedarin | 187.63 | a | 45.45 | a | 11.53 | a | 0.69 | b | 17.25 | a | 2.13 | a |
| TSKC × CTCM-008 citrangedarin | 173.79 | b | 44.39 | b | 12.04 | a | 0.72 | a | 17.21 | a | 2.18 | a |
| TSKC × CTQT 1439-003 citrangequatandarin | 190.14 | a | 46.70 | a | 11.73 | a | 0.73 | a | 16.23 | b | 2.24 | a |
| TSKC × CTQT 1439-014 citrangequatandarin | 192.23 | a | 44.33 | b | 10.56 | b | 0.58 | b | 18.49 | a | 1.93 | b |
| TSKC × CTSW-017 citrumelandarin | 173.80 | b | 42.90 | b | 11.38 | a | 0.68 | b | 17.16 | a | 2.00 | b |
| TSKC × CTSW-018 citrumelandarin | 188.18 | a | 42.60 | b | 11.08 | b | 0.74 | a | 15.09 | b | 1.92 | b |
| TSKC × CTSW-022 citrumelandarin | 178.44 | b | 46.76 | a | 11.38 | a | 0.59 | b | 20.01 | a | 2.16 | a |
| TSKC × CTSW-025 citrumelandarin | 187.60 | a | 46.95 | a | 11.90 | a | 0.76 | a | 16.43 | b | 2.29 | a |
| TSKC × CTSW-031 citrumelandarin | 184.88 | a | 44.52 | b | 11.81 | a | 0.71 | a | 17.07 | a | 2.09 | b |
| TSKC × CTSW-036 citrumelandarin | 181.77 | b | 45.37 | a | 11.05 | b | 0.67 | b | 16.74 | b | 2.06 | b |
| TSKC × CTSW-053 citrumelandarin | 187.13 | a | 43.34 | b | 10.95 | b | 0.62 | b | 17.95 | a | 1.94 | b |
| TSKC × CTSW-057 citrumelandarin | 163.33 | b | 46.81 | a | 11.63 | a | 0.75 | a | 15.95 | b | 2.21 | a |
| TSKC × CTSW-058 citrumelandarin | 172.72 | b | 44.87 | a | 11.39 | a | 0.70 | a | 16.63 | b | 2.08 | b |
| TSKC × CTTR-012 citrangedarin | 181.63 | b | 44.91 | a | 11.24 | b | 0.66 | b | 17.81 | a | 2.08 | b |
| TSKC × CTTR-028 citrangedarin | 192.04 | a | 42.36 | b | 11.26 | b | 0.68 | b | 17.19 | a | 1.96 | b |
| TSKC × CTTR-029 citrangedarin | 198.93 | a | 42.61 | b | 11.16 | b | 0.67 | b | 17.55 | a | 1.96 | b |
| TSKC × LHA-007 tangor | 186.45 | a | 46.01 | a | 11.36 | a | 0.79 | a | 14.73 | b | 2.15 | a |
| TSKC × TRBK-006 citrandarin | 192.90 | a | 45.92 | a | 11.57 | a | 0.67 | b | 17.64 | a | 2.16 | a |
| TSKC × TRFD-003 citrandarin | 196.02 | a | 45.79 | a | 11.21 | b | 0.65 | b | 17.37 | a | 2.10 | b |
| TSKC × TRFD-006 citrandarin | 174.60 | b | 47.77 | a | 11.67 | a | 0.72 | a | 17.18 | a | 2.32 | a |
| TSKC × TRFD-007 citrandarin | 191.48 | a | 42.49 | b | 11.71 | a | 0.70 | a | 17.27 | a | 2.04 | b |
| TSKFL × CTARG-023 citrangedarin | 196.27 | a | 44.45 | b | 11.05 | b | 0.70 | a | 16.10 | b | 2.01 | b |

**Table 4.** *Cont.*

| Rootstock | FW (g) | | JC (%) | | SS (°Brix) | | TA (%) | | MI (SS/TA) | | TI (kg SS box$^{-1}$) | |
|---|---|---|---|---|---|---|---|---|---|---|---|---|
| TSKFL × CTARG-029 citrangedarin | 176.90 | b | 47.77 | a | 11.22 | b | 0.64 | b | 17.88 | a | 2.19 | a |
| TSKFL × CTC13-005 citrangedarin | 176.53 | b | 44.79 | a | 11.21 | b | 0.71 | a | 16.15 | b | 2.06 | b |
| TSKFL × CTC13-012 citrangedarin | 186.13 | a | 39.15 | c | 11.38 | a | 0.65 | b | 18.00 | a | 1.83 | b |
| TSKFL × CTSW-004 citrumelandarin | 197.96 | a | 44.30 | b | 11.55 | a | 0.68 | b | 17.38 | a | 2.08 | b |
| TSKFL × CTSW-009 citrumelandarin | 180.87 | b | 44.54 | b | 11.47 | a | 0.72 | a | 16.63 | b | 2.08 | b |
| TSKFL × CTTR-004 citrangedarin | 171.18 | b | 43.51 | b | 11.42 | a | 0.70 | a | 16.58 | b | 2.04 | b |
| TSKFL × CTTR-006 citrangedarin | 153.81 | b | 43.69 | b | 11.39 | a | 0.69 | b | 17.54 | a | 2.10 | b |
| TSKFL × CTTR-013 citrangedarin | 186.36 | a | 46.73 | a | 11.20 | b | 0.69 | b | 16.65 | b | 2.13 | a |
| TSKFL × CTTR-017 citrangedarin | 185.62 | a | 46.60 | a | 11.55 | a | 0.67 | b | 18.18 | a | 2.21 | a |
| TSKFL × CTTR-021 citrangedarin | 172.51 | b | 46.20 | a | 11.42 | a | 0.72 | a | 16.16 | b | 2.14 | a |
| TSKFL × CTTR-022 citrangedarin | 197.04 | a | 46.68 | a | 11.23 | b | 0.66 | b | 17.51 | a | 2.15 | a |
| Tropical Sunki mandarin | 211.47 | a | 48.09 | a | 11.52 | a | 0.69 | b | 16.44 | b | 2.18 | a |
| F | * | | * | | * | | * | | * | | * | |
| CV (%) | 5.27 | | 3.87 | | 3.38 | | 6.46 | | 5.54 | | 4.99 | |
| Mean | 184.58 | | 45.25 | | 11.35 | | 0.69 | | 16.93 | | 2.09 | |

Means followed by the same letters in the column belong to the same group by the Scott–Knott test. (ns) not significant, (*) significant at $p<0.01$. Hybrid rootstock acronyms are described in Table 1.

For juice quality variables, two groups were always formed. Overall, in the 2010–2015 period, soluble solids concentration (SS), titratable acidity (TA), and the maturity index (SS/TA) were within the standard range of fruit harvested for processing not from concentrate (NFC) juice [51,52]. However, the technological index (TI) was slightly lower due to the juice extraction method. Ranging from 11.36 to 12.33 °Brix, 45% of rootstocks, mainly less vigorous ones, clearly induced higher SS than the Santa Cruz Rangpur lime and other vigorous rootstocks, except for the Tropical Sunki mandarin. The Santa Cruz Rangpur lime, in turn, led to lower mean TA (0.63% in October/November) alongside 60% of the evaluated rootstocks. TSKC × TRFD-006 and 007 citrandarins, TSKC × CTCM-008 citrangedarin, and TSKC × CTSW-031 citrumelandarin can be underlined as having outstanding potential for NFC due to their well-balanced SS/TA juice ratio. Finally, the Santa Cruz Rangpur lime was grouped within the lowest TI range (1.83 to 2.10 kg SS box$^{-1}$), whereas Sunki Tropical mandarin and 19 hybrid rootstocks induced higher TI (2.12 to 2.32 kg SS box$^{-1}$). Moreover, the dwarfing TSKC × TRFD-006 citrandarin stood out for the fruit quality among its full siblings because it was the only one that outperformed Santa Cruz Rangpur lime by inducing higher means of TI, SS, TA, and MI in the Valencia fruits.

*3.5. Multivariate Analyses*

There was genetic variation among the evaluated rootstocks, with the two principal components responsible for 69% of the total variation. PC1 was mainly explained by the plant height, fruit yield (kg of fruits per tree), and SS, whereas PC2 was explained by the tolerance to drought and production efficiency (kg of fruits per cubic meter of canopy) (Figure 3A). Production efficiency was positively related to the fruit yield and drought tolerance, but conversely related to the tree height, while SS was opposed to the tree height and production, juice content, and drought tolerance (Figure 3B). The cophenetic correlation was significant (r = 0.72**), and six groups of rootstocks were formed according to the similarity patterns (Figure 3B,C).

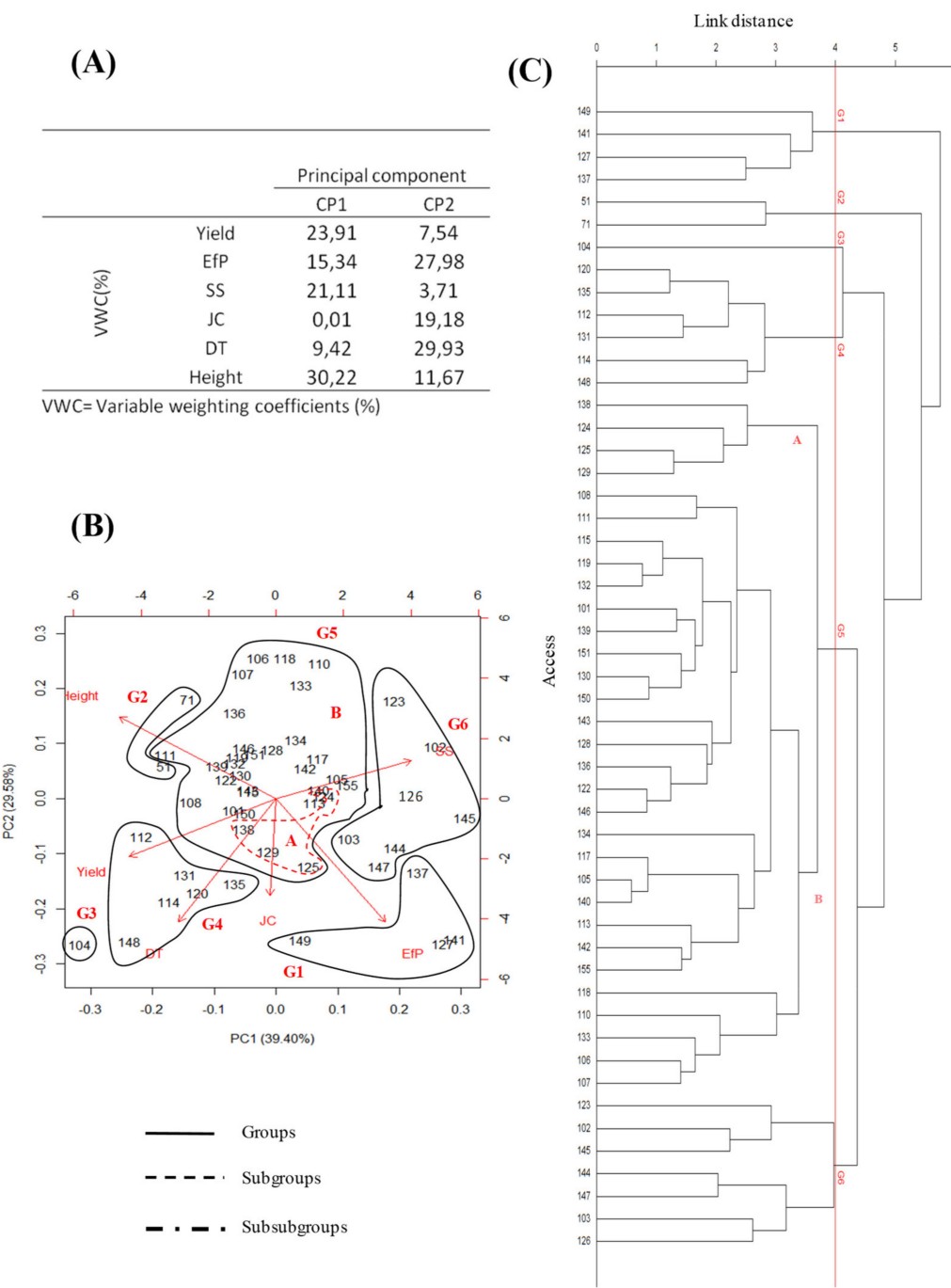

**Figure 3.** Weighting coefficients of variables (**A**), scores of the Principal Component Analysis (**B**), and dendogram (**C**), obtained by the Unweighted Pair Group Method using Arithmetic averages (UPGMA) method using the dissimilarity measurements of the following variables: fruit yield (*FY*), production efficiency (*EfP*), juice content (JC), tolerance to drought based on visual assessment (DT), plant height (PH) and soluble solids concentration in the juice (SS) of Valencia sweet orange (*Citrus ×sinensis* (L.) Osbeck) grafted onto 51 hybrid rootstocks from 2010 to 2015 in the Northern of São Paulo State, Brazil. Cophenetic correlation coefficient (CCC) = 0.72**. G1: TSKC × TRFD-003 (149), TSKC × TRFD-006 (141), TSKC × CTSW-058 (127), and TSKC × TRFD-007 (137); G2: Santa Cruz Rangpur lime (51) and Tropical Sunki mandarin (71); G3: HTR-208 (104); G4: LCR × LRF-034 (120), TSKFL × CTARG-029 (135), MXWL × LHA-001 (112), TSKC × (LCR × TR)-040 (131), TSKC × CTQT 1439-014 (114), and LCR × CTSW-009 (148); G5A: TSKFL × CTSW-004 (138), LRF × (LCR × TR)-004 (124), TSKC × CTSW-025 (125), and TSKC × CTQT 1439-003 (129); G5B: HTR-206 (108), TSKC × CTARG-020 (111), TSKFL × CTARG-023 (115), TSKC × CTSW-053 (119), TSKC × (LCR × TR)-020 (132), TSKC × CTSW-036 (101), TSKC × CTARG-019 (139), TSKC × CTTR-012 (151), LVK × CTSW-009 (130), TSKFL × CTTR-013 (150), TSKC × CTARG-015 (143)

, TSKC × CTSW-057 (128), TSKFL × CTSW-009 (136), TSKC × LHA-007 (122), TSKC × CTARG-081 (146), TSKC × CTARG-069 (134), TSKFL × CTTR-021 (117), TSKC × CTSW-022 (105), TSKFL × CTTR-022 (140), TSKFL × CTC13-005 (113), TSKC × CTTR-029 (142), Sunki × Alemow (155), TSKFL × CTC13-012 (118), TSKC × CTTR-028 (110), TSKFL × CTTR-004 (133), TSKC × CTSW-017 (106), and TSKC × CTSW-018 (107); G6: TSKFL × CTTR-006 (123), HTR-207 (102), TSKC × CTCM-008 (145), TSKFL × CTTR-017 (144), TSKC × TRBK-006 (147), TSKC × (LCR × TR)-016 (103), and TSKC × CTSW-031 (126). Hybrid rootstock acronyms are described in Table 1.

G1 (efficient dwarfing rootstocks) comprised all Sunki x Flying Dragon citrandarins plus TSKC × CTSW-058. Rootstocks within this group induced relatively high fruit yield, very high production efficiency, small tree size, and high fruit quality. The TSKC × TRFD-003 citrandarin was the only one with tolerance to drought similar to the Santa Cruz Rangpur lime, which was in the G2 (standard rootstocks) group with Tropical Sunki mandarin, being vigorous, drought-tolerant, and productive yet less efficient rootstocks. G3 (heavy-bearing rootstock) included only the HTR-208 citrandarin, which was by far the most productive rootstock in the 2010–2015 period, inducing large tree size and high tolerance to drought, but a lower concentration of SS. G4 (vigorous drought-tolerant rootstocks), with six hybrids, led to usually large trees, high fruit yield, and high tolerance to drought, but low to medium fruit quality. G5 was subdivided to facilitate interpretation, as follows: G5A (high fruit quality-inducing rootstocks), grouping four hybrids that induced high production of high-quality fruits, albeit tree size was large, resulting in low production efficiency, and the tolerance to drought was mainly low; and G5B (drought-sensitive rootstocks), comprising 27 hybrids that were sensitive to drought and induced medium to high tree size, low production efficiency, medium to low fruit yield, and good fruit quality. Finally, G6 (low-bearing dwarfing rootstocks) was characterized by decreased tree size, but the production efficiency was lower. Seven hybrids were within this group, which also induced high quality, but they were mainly more sensitive to drought and yielded less fruit.

The fruit yield and tree size were evaluated in 2018–2020 to confirm the performance of 14 selected rootstocks in the longer term (Table 5). Most vigorous and semi/dwarfing rootstocks that were previously classified as more promising confirmed their good production and similar tree size class at 12 years old, indicating that the multivariate analyses on data from initial six harvest crops allowed a consistent selection of superior rootstocks. TSKC × TRFD-006 and HTR-208 citrandarins and the Santa Cruz Rangpur lime corroborated to some tree decline, even though tree mortality was very close to that in 2018 (Table 2).

**Table 5.** Tree canopy diameter (*PD*), height (PH), and volume (*CV*) in 2020, and fruit yield (*FY*) in 2018, 2019, and 2020 of Valencia sweet orange (*Citrus×sinensis* (L.) Osbeck) on selected rootstocks in Northern São Paulo State, Brazil.

| Rootstock | *PD* | | PH | | *CV* (m³) | | *FY* 2018 | | *FY* 2019 [1] | *FY* 2020 | |
|---|---|---|---|---|---|---|---|---|---|---|---|
| | ----- (m) ----- | | | | | | ----------- (kg·tree⁻¹)----------- | | | | |
| HTR-206 citrangedarin | 5.03 | a | 4.27 | a | 57.21 | a | 96.00 | a | - | 40.87 | a |
| HTR-208 citrandarin | 4.43 | a | 2.60 | b | 26.53 | c | 103.50 | a | 13.90 | 41.22 | a |
| LCR × CTSW-009 citrumelimonia | 4.60 | a | 3.93 | a | 43.53 | b | 104.00 | a | - | 64.86 | a |
| LCR × LRF-034 lemon | 4.33 | a | 3.63 | a | 35.91 | b | 93.33 | a | - | 54.80 | a |
| Santa Cruz Rangpur lime | 4.03 | b | 3.60 | a | 30.88 | c | 48.67 | b | - | 38.05 | a |
| TSKC × (LCR × TR)-040 citrimoniandarin | 3.90 | b | 3.30 | a | 26.26 | c | 49.67 | b | - | 58.13 | a |
| TSKC × CTQT 1439-003 citrangequatandarin | 4.10 | b | 3.20 | a | 28.14 | c | 71.58 | a | 18.00 | 56.15 | a |
| TSKC × CTQT 1439-014 citrangequatandarin | 4.10 | b | 3.00 | b | 27.06 | c | 64.00 | b | - | 69.86 | a |
| TSKC × CTSW- 025 citrumelandarin | 3.10 | d | 2.53 | b | 12.91 | d | 81.00 | a | - | 27.20 | b |
| TSKC × TRBK-006 citrandarin | 3.63 | c | 2.60 | b | 18.00 | d | 53.67 | b | 7.80 | 23.00 | b |
| TSKC × TRFD-003 citrandarin | 3.70 | c | 3.06 | b | 22.94 | c | 40.00 | b | 22.70 | 47.53 | a |
| TSKC × TRFD-006 citrandarin | 2.53 | d | 1.80 | c | 6.08 | d | 15.67 | b | 9.40 | 8.80 | b |
| TSKC × TRFD-007 citrandarin | 3.10 | d | 2.20 | c | 11.08 | d | 52.83 | b | 42.80 | 44.86 | a |

**Table 5.** *Cont.*

| Rootstock | PD | | PH | | CV (m³) | | FY 2018 | | FY 2019 [1] | FY 2020 | |
|---|---|---|---|---|---|---|---|---|---|---|---|
| | ----- (m) ----- | | | | | | ----------- (kg·tree⁻¹)----------- | | | | |
| TSKFL × CTARG-029 citrangedarin | 3.87 | b | 3.63 | a | 28.31 | c | 97.67 | a | - | 46.33 | a |
| TSKFL × CTSW-004 citrumelandarin | 4.83 | a | 3.63 | a | 44.53 | b | 73.00 | a | - | 62.55 | a |
| Tropical Sunki mandarin | 4.40 | a | 4.03 | a | 40.88 | b | 50.67 | b | 15.30 | 26.75 | b |
| F | * | | * | | * | | * | | - | * | |
| *CV* (%) | 9.20 | | 13.19 | | 25.36 | | 29.40 | | - | 44.78 | |
| Mean | 3.98 | | 3.18 | | 28.76 | | 66.12 | | - | 44.43 | |

Means followed by the same letters in the column belong to the same group by the Scott–Knott test. (ns) not significant, (*) significant at $p < 0.01$. Hybrid rootstock acronyms are described in Table 1.[1] In 2019, fruit yield was very low due to severe drought conditions, and therefore data were not statistically analyzed. (-) not evaluated due to insignificant fruit load on the tree.

## 4. Discussion

Tree architecture plays a determinant role for productivity because it regulates several components of the production, from photosynthesis (e.g., light intercepting, harvest index) to crop management (e.g., plant arrangement, cultural practices) [53]. The density of trees per unit area has grown over the years, which implies the use of smaller trees (smaller height, diameter, and canopy volume) to facilitate cultural practices (pruning, pest and disease control, fertilization, fruit harvesting, etc.) [54]. Therefore, manipulation of how trees grow has been a major goal of breeding and cropping systems [55–57]. This is particularly critical for fruit crops, which are, in general, perennial woody species, and hence challenges the sustainability of both fresh fruit and processing in modern horticulture [58].

The use of dwarfing rootstocks has proven to be one of the most efficient methods to control tree size and enhance the productivity and has been developed to some extent for several fruit crops including citrus [10,59,60]. Herein, we report 17 citrus hybrid rootstocks that induced tree size reduction in the Valencia sweet orange from 40% to 75% compared with standard rootstocks in Brazil. Although other hybrid rootstocks that moderately decrease the scion canopy volume are available [17,22,61], only the Flying Dragon trifoliate orange can be considered truly dwarfing in citriculture so far, alongside a few canopy varieties such as Satsuma mandarin (*C. unshiu* Marcow.) and Fuya Meñuda sweet orange [18,62]. The selected hybrids presented in this work may potentially allow a wider genetic diversification in orchards that require smaller trees to better adjust to high tree density, disease management, and mechanical harvesting [12,63].

The genetic control of tree size by citrus dwarfing varieties has not been fully understood, and different physiological mechanisms have been reported such as hormonal balance with a higher relation of abscisic acid (ABA) to auxins and gibberellins [64,65], carbohydrate partitioning and sink–source relations [61,66], tissue affinity at the graft union [24,67], smaller vessel elements in the xylem [68], and lower hydraulic conductance [69,70]. The *rol*ABC and a *GA20-oxidase* genes have been related to the expression of the dwarfism phenotype in citrus [71,72]. In this work, all, but not only, Sunki mandarin × Flying Dragon trifoliate orange citrandarins were semi or dwarfing, with ten different parental crosses generating less vigorous rootstocks. Although a genetic background with *Poncirus trifoliata* as a male genitor is usually a rule [73], dwarfing progenies have been previously reported for crosses using different species, even for intraspecific hybrids of vigorous lemon types [74]. The dwarfism character of the heterozygous Flying Dragon genotype seems to derive from a mutation, and despite being inheritable as a single dominant gene, a 5:1 segregation pattern for tree size control was observed for progenies from its self-cross [75]. As a consequence of the horticultural importance of this trait, it is expected that major efforts to broaden the knowledge on the genetic regulation of citrus dwarfism may be dispended in the short-term, aiming at a faster and more precise advancement in obtaining improved varieties, notably rootstock, as it was observed for other fruit crops [76].

The evaluated semi-dwarfing and dwarfing rootstocks were grouped according to their fruit yield and production efficiency. Among them, only the Sunki × Flying Dragon citrandarins led to higher production without presenting unequivocal symptoms of graft incompatibility. Although some trees on TSKC×TRFD-007 presented a marked line at the grafted union, tree stand was healthy and uniform, and trees were very productive. These same hybrids had previously demonstrated additional advantages in propagation due to high polyembryony rates and high vigor in the greenhouse [77], which are issues with the Flying Dragon trifoliate orange [73]. The majority of dwarfing rootstocks were sensitive to drought, which may explain the lower yields yet high concentrations of soluble solids, considering that the crop was rainfed under cyclic drought seasons. The performance of the Flying Dragon trifoliate orange, that is, the final tree size and fruit production, is largely influenced by the cropping system, environmental conditions, use of irrigation, and different scion varieties [19,78,79]. All these factors should be taken into account for the commercial use of any dwarfing rootstock. Therefore, all selected hybrids in this work are worthy of further evaluation to identify other potential graft combinations. Even unexpected loss of the dwarfing character and other limitations regarding resistance to biotic and tolerance to abiotic stresses may be clarified after long-term studies, since tree size decrease may result from environmental constraints rather than only genetic control [80]. In this work, in-rows tree spacing used (2.5 m) was smaller than the canopy width induced by most rootstocks at bearing age, which may have limited tree growth to some extent and affected fruit yield and quality. However, the differences among the evaluated rootstocks were clearly related to the genotypes over the assessment period, and the horticultural evaluation was carried out under uniform conditions that are representative of commercial management, therefore being practical for citrus growers.

Tree architecture in addition to carbohydrate competition and hormonal control may also explain the better performance of some dwarfing hybrids because production efficiency and fruit quality but not drought tolerance were related to the canopy volume. The production of lots of high-quality fruit per cubic meter of the canopy is a frequent trait of the Flying Dragon as well [20]. A suitable tree spacing for ultra-high-density citrus orchards in Brazil could be recommended as a function of the canopy equatorial diameter (D) at adult age, with D+ 2.0 and D x 0.65, in meters, for the between and in-rows spacing, respectively, adapted from De Negri [81]. Considering this calculation and the tree size at eight years old induced by one dwarfing (TSKC × TRFD-006), one semi-dwarfing (TSKC × TRFD-003), one standard (Santa Cruz Rangpur lime), and one vigorous (HTR-208) rootstock, the estimated tree density would be of 1429, 1100, 704, and 635 trees·ha$^{-1}$, respectively. The estimated fruit productivity, considering the mean fruit yield in the 2010–2015 period and ignoring any significant reduction in this variable due to the closer spacing [15], would result in 45.9, 43.3, 27.8, and 43.3 t·ha$^{-1}$, respectively. In 2019, the highest mean fruit productivity in the historical series of São Paulo State was registered, at 42.6 t·ha$^{-1}$ [7]. It is clear that using closer tree spacings, that is, adequately arranging the highly efficient canopy in the provided space, is mandatory for semi-dwarfing or dwarfing rootstocks, because a 2-fold higher yield would be expected in relation to the productivity obtained at the current tree density, 666 trees·ha$^{-1}$. This suggests that semi-dwarfing or dwarfing rootstocks may present potential high yield in ultra-high-density orchards in order to compete with vigorous heavy-bearing rootstocks at wider tree spacing. Moreover, only the less vigorous rootstocks would allow functionalities other than yield, for instance, reduced need for pruning, easy plant care, and most significant of all, the feasibility of mechanical harvesting. Furthermore, fruit quality on dwarfing hybrids was usually higher, reinforcing that this class of rootstock will likely be more suitable for a sustainable citrus industry provided that high productivity is addressed.

Although the objective in this work was to select dwarfing rootstocks, some vigorous ones should be underlined. The HTR-208 citrandarin induced an outstanding higher fruit yield, and the six hybrids in the G4 group (LCR × LRF-034, TSKFL × CTARG-029, MXWL × LHA-001, TSKC × (LCR × TR)-040, TSKC × CTQT 1439-014, and LCR × CTSW-

009) produced similar yields to the commercial rootstocks, Tropical Sunki mandarin and Santa Cruz Rangpur lime. Their tolerance to drought was very high, which may justify their cultivation without irrigation even though fruit internal quality was not the best for pasteurized juice processing. Additionally, the tree size of selected rootstocks was measured again in 2020, and although there was some variation in the canopy volume, notably for the more vigorous rootstocks, trees started to be managed by pruning after ten years of planting to limit tree growth to the tree spacing. On the other hand, HTR-206 citrangedarin resulted in excessive tree size.

Based on the results of this experiment and results from other validation areas in different regions of Brazil, some hybrid rootstocks were selected and have been registered at the Brazilian Ministry of Agriculture, Livestock and Food Supply for release to growers (Supplementary Materials ).

## 5. Conclusions

Seventeen citrus hybrids were selected as semi-dwarfing and dwarfing rootstocks in relation to the standards, Santa Cruz Rangpur lime and Sunki Tropical mandarin, with TSKC × TRFD-003, 006, and 007 citrandarins standing out for inducing higher fruit yield and quality in the Valencia sweet orange scion variety. The first hybrid is highlighted due to a higher tolerance to drought compared with its siblings, and particularly TSKC× TRFD-006 should be considered for irrigated crop. These citrandarins present potential for cultivation under tropical and subtropical climates, and preferably in irrigated high-density orchards, in addition to the vigorous HTR-208 and LCR × CTSW-009 hybrids, because of their superlative fruit yield.

**Supplementary Materials:** The following are available online at https://www.mdpi.com/article/10.3390/agriculture11040354/s1, Supplementary Material 1: Rootstocks obtained by the Citrus Breeding Program of Embrapa Cassava & Fruits registered or under registration in the Ministry of Agriculture, Livestock and Food Supply, Brazil.

**Author Contributions:** Conceptualization, E.S.S. and W.d.S.S.F.; data verification, D.P.C., C.A.d.S.L., E.A.G., E.S.S., and A.S.M.; writing and editing, D.P.C., E.A.G., E.S.S., W.d.S.S.F., and C.A.d.S.L.; review, D.P.C., E.A.G., E.S.S., W.d.S.S.F., A.S.M., C.A.d.S.L., O.S.P., A.d.S.G., M.A.C.F., A.L.V.S., and H.C.d.L.; acquisition of financing, A.d.S.G. and W.d.S.S.F.; technical support, H.C.d.L. and A.L.V.S. All authors have read and agreed to the published version of the manuscript.

**Funding:** Embrapa Mandioca e Fruticultura (Processes 02.13.03.005.00.00 and 20.18.01.007.00.00), Coordenação de Aperfeiçoamento de Pessoal de Nível Superior (Financial Code 001) and Conselho Nacional de Desenvolvimento Científico e Tecnológico (Process 425395/2016-2) provided financial aid.

**Institutional Review Board Statement:** Not applicable.

**Informed Consent Statement:** Not applicable.

**Data Availability Statement:** The dataset for this research is available upon reasonable request to the corresponding author.

**Acknowledgments:** To the Coordenação de Aperfeiçoamento de Pessoal de Nível Superior (CAPES)-Financial Code 001, for the doctorate fellowship to D.P.C.; to the Conselho Nacional de Desenvolvimento Científico e Tecnológico (CNPq), for the grants to E.S.S., A.d.S.G., M.A.C.F., and W.d.S.S.F.; to Citrosuco S/A Agroindústria and Muriti farm, for the experimental area and technical support; to the Fundação Coopercitrus-Credicitrus (FCC), for technical support; to Embrapa Mandioca e Fruticultura, for the plant material and financial aid (Processes 02.13.03.005.00.00 and 20.18.01.007.00.00); to Antônio Santana da Silva, Getúlio de Souza Vieira, and Magno Guimarães Santos, for technical support; and to Otávio Ricardo Sempionato, for suggestions regarding this work.

**Conflicts of Interest:** The authors declare no conflict of interest.

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
