# Peer review of "Less Is More: A Hard Way to Get Potential Dwarfing Hybrid Rootstocks for Valencia Sweet Orange"

_agriculture, doi:10.3390/agriculture11040354_

Round 1

Reviewer 1 Report

This submission presents information from a long-term rootstock trial planted at moderately high density in SP, Brasil. These types of submissions are always difficult to judge since there are so many variables and usually they do not separate out as nicely as wished. In this case, the authors conclude that three dwarfing rootstocks producing good fruit quality and yield were worthy of inclusion in the national program, along with two vigorous hybrids. Although not real exciting, these findings will be of interest to some readers and are worthy of publication after some minor editing, mostly involving English usage. The paper is easily understood as it stands but will be better after cleaning up the English, particularly with respect to conjunctions. For instance, the authors consistently use "concerning" when "compared to" is needed. These small things are the most difficult in non-native languages (as I know from my own experience!).

The language used to describe the amount of dwarfing was unclear in lines 26 - 27, 55 - 56, and 76 - 77. This was clear in 139 - 143, but the earlier references to tree size reduction should be clarified. 

180 - 181: This means (dead trees of genotype A)/(total planted trees of genotype A), correct? If so, suggest re-writing so that it is clear that this is per genotype.
263 - 279: This paragraph belongs in section 3.2.
275 - 279: Please re-write; it is confusing as written.
356: "Heavier": do you mean "large"? Or "Heavier than those produced by Santa Cruz rangpur"?
383: "german brothers": meaning unknown in English.
552 - 555 and 558  - 565: Non dwarfing rootstocks are shown with names in supplementary material, but names are not given to the actual dwarfing rootstocks.
561: "Former" should be "first".

So, clean this up a bit and you are good to go.

Author Response

All the reviewers' suggestions were taken care of.

Reviewer 2 Report

The study is of high relevance to the field of citriculture, especially rainfed conditions. The study executed well and good information generated from field experimentation over a long period. However, there is some concern noticed, while recommending the rootstocks for specific purposes and conditions, highlighted in the attached file of the manuscript. So, authors must revisit the result data before giving the recommendation of the rootstocks for specific purposes looking from all angles to avoid any discrepancies. This has been highlighted in the abstract, discussion, and finally in conclusion. In addition, there are some typo and syntax errors in the texts that need to address in the revised form.

Bset

Author Response

(The authors gave the same response as above.)

Reviewer 3 Report

The authors have presented the results of a comprehensive study on sweet orange dwarfing hybrid rootstocks including a considerable number of hybrids.

I support the publication of the work in its current form.

Just a minor modification that I may recommend would be the addition of some imagery for example from the ’’the presence of a fine line between the scion and the rootstock tissues’’ or other visual effects that have been detected during this study. Also a photo collection of the selected seventeen citrus hybrids trees and their representative fruits would attract the reader’s attention more. The latter can be presented as supplementary materials.

Line 178: Change ‘’water stress’’ to ‘’water shortage stress’’ or ‘’drought stress’’

Author Response

All the reviewers' suggestions were taken care of.

Unfortunately, at moment we do not have photographs of the plants and fruits of the selected hybrid rootstocks.
